# EEG *p*-adic quantum potential accurately identifies depression, schizophrenia and cognitive decline

**Oded Shor**[1,2], **Amir Glik**[2,3,4], **Amit Yaniv-Rosenfeld**[2¤], **Avi Valevski**[2,5],
**Abraham Weizman**[1,2,5], **Andrei Khrennikov**[6☉], **Felix Benninger**[1,2,3☉] *

**1** Felsenstein Medical Research Center, Petach Tikva, Israel, **2** Sackler Faculty of Medicine, Tel Aviv University, Tel Aviv, Israel, **3** Department of Neurology, Rabin Medical Center, Petach Tikva, Israel, **4** Cognitive Neurology Clinic, Rabin Medical Center, Petach Tikva, Israel, **5** Geha Mental Health Center, Petach Tikva, Israel, **6** Faculty of Technology, Department of Mathematics Linnaeus University, Växjö, Sweden

☉ These authors contributed equally to this work.
¤ Current address: Shalvata Mental Health Center, Hod Hasharon, Israel
* benninger@tauex.tau.ac.il

**Data Availability Statement:** Anonymized retrospective EEG data is available as a three-dimensional matrix as a matlab file at https://

## Abstract

No diagnostic or predictive instruments to help with early diagnosis and timely therapeutic intervention are available as yet for most neuro-psychiatric disorders. A quantum potential mean and variability score (qpmvs), to identify neuropsychiatric and neurocognitive disorders with high accuracy, based on routine EEG recordings, was developed. Information processing in the brain is assumed to involve integration of neuronal activity in various areas of the brain. Thus, the presumed quantum-like structure allows quantification of connectivity as a function of space and time (locality) as well as of instantaneous quantum-like effects in information space (non-locality). EEG signals reflect the holistic (nonseparable) function of the brain, including the highly ordered hierarchy of the brain, expressed by the quantum potential according to Bohmian mechanics, combined with dendrogram representation of data and *p*-adic numbers. Participants consisted of 230 participants including 28 with major depression, 42 with schizophrenia, 65 with cognitive impairment, and 95 controls. Routine EEG recordings were used for the calculation of qpmvs based on ultrametric analyses, closely coupled with *p*-adic numbers and quantum theory. Based on area under the curve, high accuracy was obtained in separating healthy controls from those diagnosed with schizophrenia (p<0.0001), depression (p<0.0001), Alzheimer's disease (AD; p<0.0001), and mild cognitive impairment (MCI; p<0.0001) as well as in differentiating participants with schizophrenia from those with depression (p<0.0001), AD (p<0.0001) or MCI (p<0.0001) and in differentiating people with depression from those with AD (p<0.0001) or MCI (p<0.0001). The novel EEG analytic algorithm (qpmvs) seems to be a useful and sufficiently accurate tool for diagnosis of neuropsychiatric and neurocognitive diseases and may be able to predict disease course and response to treatment.

datadryad.org/stash/share/AWmC0-
Afzx29cOkYDXQ6y2-7HF4GBvG-J-9i8hDQZsw.

**Funding:** The author(s) received no specific
funding for this work.

**Competing interests:** The authors have declared
that no competing interests exist.

## Introduction

Disorders of the brain, such as schizophrenia, epilepsy, depression and dementia, constitute approximately 27% of the global disease burden in terms of disability-adjusted life-years (DALYs) and that surpasses cardiovascular diseases and cancer combined [1]. For most brain disorders no single accurate, diagnostic tool is available as yet [2–4]. Several biomarkers exist to substantiate the diagnosis, including biological markers from serum or cerebrospinal fluid (CSF), neuroimaging techniques, including magnetic resonance imaging (MRI), functional MRI (fMRI), and positron emission tomography (PET). Those, however, are often expensive possibly quite invasive and none of these techniques has yielded a biomarker sufficient for accurate diagnosis of disorders such as Alzheimer's disease (AD) [5], major depression or schizophrenia [6]. Electroencephalography (EEG) is an inexpensive and well-established tool [7, 8] used for resting-state power, spectral and functional connectivity analyses as well as microstate analysis, which may assist in diagnosing these disorders [9–14] with variable success and little use in clinical practice.

The recent years were characterized by tremendous development of quantum information theory [15–17]. Nowadays quantum-like modelling is widely used in microbiology, genetics, cognition, psychology, decision making and social science [18–22]. Furthermore, it is widely accepted that the brain that is considered a "black box" in this model, is a highly hierarchic organ in terms of communication and subsystem function [23, 24]. Baring this in mind, we decided to compare the EEG-pattern of healthy controls and those of peoples with neuropsychiatric diseases. The way to represent hierarchy in mathematical terms is by dendrogram trees that can be expressed as $p$-adic numbers, [21, 22, 25] representing an emergent property of the holistic brain (where throughout this study the 2-adic numbers are in the use thus $p$ = 2). The quantum (Bohmian mechanics) formalism was used operationally to describe holistic information processing by the brain in accordance with Bohr and Hilley who treated it as a field of "active information" [26].

The current study's main objective was to develop a novel and relatively simple tool to diagnose and predict multiple neuropsychiatric diseases. This tool combines routine EEG and mathematical structures of quantum Bohmian theory, to extract characteristic information patterns presented in dendrograms, expressing the hierarchical treelike structure of information processing in the brain [27]. This novel method accurately identified participants with mild cognitive impairment (MCI), AD, schizophrenia, or depression, by routine EEG records analysed by this novel approach.

## Methods

The study adhered to rules and regulations of the Helsinki Declaration and was approved by the Institutional Review Board (IRB) of the Rabin Medical Center, Petach Tikva, Israel (0275-20-RMC). The study was approved as retrospective clinical and need for consent was waived by the ethics committee. All patient data were fully anonymized before review.

### Participant groups

Online medical health records from two medical centres were used to identify all participants that underwent at least one routine EEG examination between the years 2011 and 2019. The participants were then divided into the following groups: 1) Depression: Participants with a diagnosis of major depressive disorder (MDD), hospitalized during the index time. This diagnosis had been established by two senior psychiatrists according to DSM-IV and DSM-V criteria, following a psychiatric interview where the severity of depression was found to be at least moderate. In addition, the participants (range: 33–91 years; average age: 69.7 ±14.8 years; 20 females) had to have had at least one previous major depressive episode, prior to age 30, namely,

the index episode was a recurrent one. 2) Schizophrenia: Diagnosis of schizophrenia had been established by two senior psychiatrists according to the ICD-10 criteria. In addition, the participant had to be hospitalized during the index time. 3) Cognitive impairment: Participants in the study had been diagnosed by two senior neurologists, with either MCI or AD according to the criteria of the National Institute on Aging and the Alzheimer's Association [28, 29]. 4) Controls: Participants undergoing routine EEG due to indications unrelated to neuropsychiatric diseases. None of the participants in this group had been diagnosed with any condition defining any of the other groups. In this group, exclusion criteria also included diagnosis of bipolar disorder; substance abuse, psychiatric or general medical conditions requiring hospitalization, history of epilepsy or conditions requiring anticonvulsants, ECT, vagal nerve stimulation, or transcranial magnetic stimulation (TMS), history of traumatic brain injury and history or imaging findings of cerebrovascular diseases including ischaemic and haemorrhagic stroke.

## EEG data acquisition and analysis

Routine EEG recordings were retrospectively obtained from the medical records of all patients. EEGs had been performed in a routine clinical setting by an experienced technician. All included participants had undergone EEG between 8 am and 1 pm using a Nihon Koden surface EEG (19-electrode standard according to the international 10–20 electrode placement system) with a sampling frequency of 500 Hz (Nihon Kohden, Japan). Participants had been resting with open and closed eyes. Those who underwent sleep-EEGs were excluded.

**2-adic quantum potential calculation.** To extract the 2-adic quantum potential from participants' EEG signals, the following procedures were preformed:

1. Raw EEG data from the 19 active electrodes (*elec*) were transformed into the European Data Format (EDF).

2. Data was filtered first to remove the 50Hz mains signal and then further filtered with a high-pass 1 Hz filter. Subsequent analysis was performed using a 351s sample of continuous EEG.

3. A moving time window of 1s duration was selected. For each time step,
   We define

   $$W = window\ duration$$

   and

   $$n = \lfloor \frac{no.of\ data\ points}{sample\ rate} \rfloor \tag{1}$$

   Thus, the time step, t, has values in the interval: $t \in [1, n/W]$ and for each electrode we assign, elec, with values in the interval as the number of EEG electrodes used to record the data.

   $$elec \in [1, 19]$$

4. normalized distribution function construction. Electrical potential values, donated as *ep* with units of [*mV*] recorded from each electrode (*elec*) at any given time step, *t*, normalized according to the following:

   $$\widehat{ep}_{elec,t} = \frac{|ep_{elec,t}|}{\max(|ep_{elec,t}|)},\ \widehat{ep}_{elec,t} \in [0\ 1] \tag{2}$$

5. 19 histograms, $h_{elec,t}$, each with bin width of 0.01 were constructed representing an empirical probability distribution function of the normalized electrical potential values of each of the 19 electrodes. Thus, we have for each $elec \in 1,2..19$ a vector with 100 elements where each element represents frequency of the corresponding binned $\widehat{ep}$ values. $h_{elec,t}$ contains 19 such vectors.

6. For each $t$, we calculated the pair-wise Hellinger distances between all the 19, $h_{elec,t}$, histograms vectors as shown below:
For $h_{elec,t}$ where elec = i elec' = j and t = 1,2,...351. we have $x = h_{i,t}$ $y = h_{j,t}$
$x = x_i$ i = 1,2,3..k, $y = y_i$ i = 1,2,3..k where k in our computation is k = 100 and the Hellinger distance between vectors x and y is defined as:

$$H(x, y) = \frac{1}{\sqrt{2}} \left( \sqrt{\sum_{i=1}^{k} (\sqrt{x_i} - \sqrt{y_i})^2} \right.$$

From all H values we constructed a dendrogram with 19 edges (each edge represents one of the 19 electrodes histograms, $h_{elec,t}$).
The resulted dendrogram is a representation of distance relations of normalized electrode voltage histograms for each time window.

7. Each dendrogram was represented in a matrix ($B_{t,participant}$), where each row ($r_{elec,t,participant}$) is the 2-adic expansion of the electrode (edge (tree route in the dendrogram [30]. Each 2-adic expansion ($r_{elec,t,participant}$) was converted to a rational number in the following manner:
Thus for each binary vector $r_{elec,t,participant}$, which is a 2-adic expansion, we have a vector $Y$ that contains values of places in the 2-adic expansion of each $r_{elec,t,participant}$ equal to 1,

$$q_{elec} = \sum 2^{-Y} \quad q_{elec} \in [0 \ 1] \tag{3}$$

8. from the 19 values in $q_{elec}$ we constructed an empirical probability distribution function (pdf), $\rho(q)$, of with a kernel function of bandwidth:

$$(\max(q_{elec}) - \min (q_{elec}))/(no.columns \ of \ B_{t,participant}) \tag{4}$$

9. The quantum potential (QP) function was calculated according to P. Holland [31]:

$$QP_{t,participant} = \frac{h^2}{4m\rho} \left( \frac{1}{2\rho} \frac{\partial \rho}{\partial q} \frac{\partial \rho}{\partial q} - \frac{\partial^2 \rho}{\partial q \partial q} \right) \tag{5}$$

in our numerical approach
We define
$d = (max(q_{elec}) - min(q_{elec}))/m$ where
$m = 100$ in our numerical calculations as the number of points in the empirical probability distribution function $\rho(q)$. Interestingly our qualitative results do not change upon increasing number of points in $\rho(q)$ from 100 to 1000 and changing m respectively from 100 to 1000. further we define

$$q \in \min(q_{elec}), \min(q_{elec}) + d, \min(q_{elec}) + 2d \ldots \max(q_{elec})$$

and

$$\frac{\partial \rho}{\partial q} = \frac{\rho(q+d) - \rho(q)}{d}, \quad \frac{\partial^2 \rho}{\partial q \partial q} = (\frac{\rho(q+2d) - \rho(q+d)}{d} - \frac{\rho(q+d) - \rho(q)}{d})/d \quad (6)$$

Thus, inserting into Eq 5

$$QP_{t,participant} = \frac{h^2}{4m\rho} (\frac{1}{2\rho} \left(\frac{\rho(q+d) - \rho(q)}{d}\right)^2 - \frac{\left(\frac{\rho(q+2d) - \rho(q+d)}{d} - \frac{\rho(q+d) - \rho(q)}{d}\right)}{d}) \quad (7)$$

with Planck's constant $h = 1$ and mass $m = 1$, and $q \in [0\ 1]$.

The integral of the QP was calculated for each $t$ and each participant as follows:

$$\int QP_{t,participant} dq \quad (8)$$

We note that Eq 7 gives as a measure of the dendrogramic hierarchical structure topology. *Thus, the ambiguous quantum potential notion becomes in our framework quite trivially a score or measure of hierarchical topology.*

For each electrode, the QP value was extracted as follows:

$$QP_{elec,t,participant} = QP_{t,participant}(elec) \quad (9)$$

We define

1. The mean of the log10 ($|QP_{elec,t,participant}|$) values across electrodes for each participant as: $Qme_{t,patient}$

2. The mean of $Qme_{t,patient}$ across all participants in one group as: $Qme_{t,group}$

3. The standard deviation (STD) of $Qme_{t,patient}$ across all participants in one group as: $Qstd_{t,group}$

The above QP time series data analysis was performed with MATLAB software (Mathworks, Natick, MA).

**Quantum potential mean and variability score (qpmvs).** To compare two participant groups, we defined the following parameters.

1. For each participant, the mean log10 of the absolute value of the integral of the QP function of all $t$ was calculated as follows:

$$mIqp_{participant} = < (log10|\int QPpdq)_t|>_{participant}$$

2. For each participant group, the mean $mIqp_{participant}$ across all participants of a group was calculated as follows:

$$mIqp_{group} = < mIqp_{participant}>_{group}$$

3. For each participant, the standard deviation (std) of the log10 of the absolute value of the integral of the QP function for all $t$ was calculated as follows:

$$sIqp_{participant} = std(log10|\int (QPpdq)_t|)_{participant}$$

4. For each participant group, the mean $sIqp_{participant}$ of each group's std as follows:

$$sIqp_{groups} = <std(log10|\int (QPpdq)_t|)_{participant}>_{groups}$$

5. The quantum potential mean and variability score (qpmvs) was calculated as follows: For each participant $M$ will be the number of time steps, t, that satisfy

$$(log10|\int (QPpdq)_t|)_{participant} > mIqp_{group} + sIqp_{groups}$$

or

$$(log10|\int (QPpdq)_t|)_{participant} < mIqp_{group} - sIqp_{groups}$$

Thus,

$$\text{qpmvs}_{participant} = mIqp_{patient} * M$$

6. Receiver operating characteristic (ROC) was used to assess the accuracy of the method in differentiating the participant groups from each other. The area under the curve (AUC) is calculated as an effective measure of accuracy using the individual qpmvs$_{participant}$ with MATLAB software scripts.

**QP power spectrum analysis of QP$_{elec}$ values.** In order to study QP time-series dynamics for each participant group, fast Fourier transformation (FFT) was used for creating a spectrogram for each QP$_{elec}$ for each participant's frequency band of $2^{-n}$ (n = 1...5) and a window of 64 s with 0.5 s overlap. Each participant's electrode spectrogram (n = 19) was averaged ($<SP_{elec}>_{participant,window}$) and averaged again across all participants in each group ($<<SP_{elec}>_{participant}>_{window}$). For each frequency band, each participant group ($<<SP_{elec}>_{participant}>_{window}$) was normalized to the corresponding maximum value of that particular band.

## Statistical analyses

Statistical analyses were performed using GraphPad Prism Software (San Diego, CA). Means were represented with standard error of means (SEM). Student-t-tests were performed to compare pairwise group differentiation with a 99% confidence level. Analysis of Variance (ANOVA) tests with multiple comparisons were applied to test differentiation of all groups with a 99% confidence level.

## Results

### Participants' characteristics

A total of 230 participants (average age: 58.2 ±18.7 years; range: 18–91 years; 129 (56.1%) female) were included in the study (Table 1). The participants were grouped according to the

**Table 1. Participants' demographics.**

| | | | N | f/m | Age (y ±SD; range) |
|---|---|---|---|---|---|
| Control | | | 95 | 63/32 | 52.2 ±16.8; 19–80 |
| Depression | | | 28 | 20/8 | 69.7 ±14.8; 33–91 |
| Schizophrenia | | | 42 | 15/27 | 41.4 ±16.8; 18–76 |
| Cognitive Decline | | | 65 | 31/34 | 72.9 ±7.2; 60–87 |
| | AD | | 40 | 20/20 | 72.7 ±7.9; 60–87 |
| | MCI | | 25 | 11/14 | 73.5 ±6.0; 62–85 |
| | | stMCI | 6 | 0/6 | 74.3 ±4.6; 67–80 |
| | | dtMCI | 9 | 6/3 | 73.2 ±5.6; 65–82 |

AD–Alzheimer's Disease; MCI–mild cognitive impairment; stMCI–stable MCI; dtMCI- deteriorating MCI; SD–standard deviation; range–range of age in years

clinical data described in the methods section and consisted of 28 participants with a primary diagnosis of MDD (average age: 69.7 ±14.8 years; range: 33–91 years; 20 (71.4%) female); 42 participants with a diagnosis of schizophrenia (average age: 41.4 ±16.8 years; range: 18–76 years; 15 (35.7%) female); 65 participants with cognitive impairment (average age: 72.9 ±7.2 years; range: 60–87 years; 31(47.7%) female) from which 25 (38.5%) were diagnosed with MCI (average age: 73.5 ±6.0 years; range: 62–85 years; 11 (44%) female) and 40 with AD (average age: 72.6 ±7.9 years; range: 60–87 years; 20 (50%) female). Further, 95 participants with no neurological or psychiatric morbidity were included in the control group (average age: 52.2 ±16.8 years; range: 19–80 years; 63 (66.3%) female).

## Characterization of neuropsychiatric participant groups according to *p*-adic quantum potential

The study aimed first to differentiate participants with neuro-psychiatric disorders from control participants. For this comparison, a cumulative distribution function (CDF) of $Qme_{t,group}$ (Fig 1A) was constructed. Among the controls (n = 95), the mean of the $Qme_{t,group}$ was 4.15 ±0.03, which differed with high statistical significance from participants with depression (n = 28; 4.26 ±0.03; p<0.001), schizophrenia (n = 42; 4.24 ±0.04; p<0.001), AD (n = 40; 4.14 ±0.03; p<0.001) and MCI (n = 25; 4.17 ±0.03; p<0.001; Fig 1B and 1C; Table 2). Interestingly, the variability across participants within each group, denoted as $Qstd_{t,group}$, also differed significantly between the control group and each neuro-psychiatric disorder group (control: n = 95, 0.16 ±0.01; depression: n = 28; 0.46 ±0.06; p<0.001; schizophrenia: n = 42; 0.39 ±0.05; p<0.001; AD: n = 40; 0.15 ±0.02; p<0.001; and MCI: n = 25; 0.16 ±0.03; p<0.001; Fig 1D–1F, Table 3). The study further intended to identify the participants' specific neuro-psychiatric disorder in accordance with the quantum-like structure of the brain. For this purpose, a comparison was done among the $Qme_{t,group}$ and all groups of participants (control, AD, MCI, depression, and schizophrenia). This enables us to use $Qme_{t,group}$ as a specific biomarker in identifying participants with different neuro-psychiatric disorders. The cumulative distribution function (CDF) of $Qme_{t,group}$ and the variability ($Qstd_{t,group}$) differentiated highly significantly between the disease groups, with the exception of the variability between AD and MCI (Fig 1 and Tables 2 and 3).

## QP cross-correlation between participant electrodes

EEG signals have traditionally been used to examine the functional cortical connectivity between different areas of the brain. Connectivity measurements using scalp recording signals,

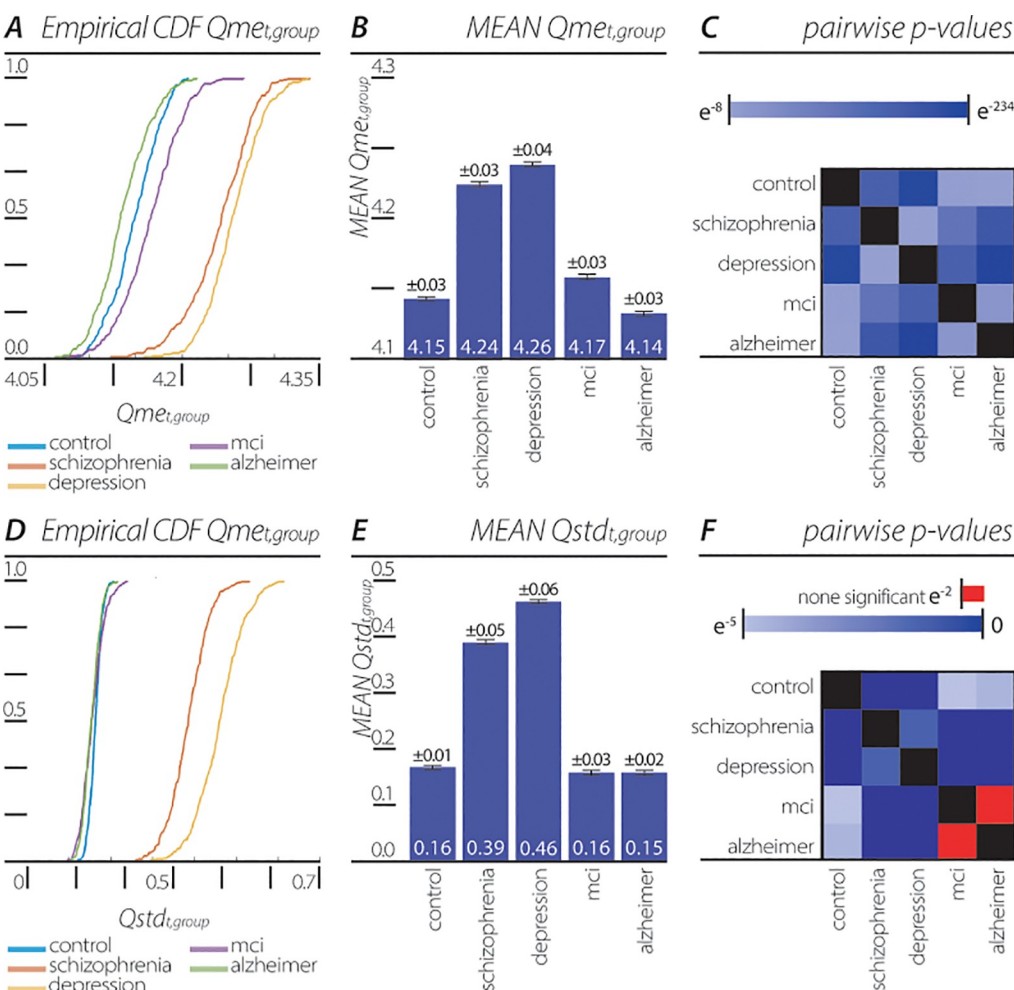

**Fig 1. Distribution of *p*-adic QP values extracted from EEG of neuro-psychiatric patient groups.** (A) Cumulative distribution function (CDF) of $Qme_{t,group}$ for control patients (n = 95), patients with depression (n = 28), schizophrenia (n = 42), AD (n = 40), and MCI (n = 25). (B) mean of $Qme_{t,group}$ the for comparison (SEM; control: 4.15 ±0.03; depression: 4.26 ±0.03; schizophrenia: 4.24 ±0.04; AD: 4.14 ±0.03; MCI:4.17 ±0.03). (C) $Qme_{t,group}$ significance p-value matrix of pairwise group comparison. (D) cumulative distribution function (CDF) of $Qstd_{t,group}$ of all five patient groups. E, mean of the $Qstd_{t,group}$ (SEM; control: 0.16 ±0.01; depression: 0.46 ±0.06; schizophrenia: 0.39 ±0.05; AD: 0.15 ±0.02; MCI: 0.16 ±0.03). F, $Qstd_{t,group}$ significance p-value matrix of pairwise group comparison.

include the Pearson coefficient of correlation, coherence, phase lag, and synchronization likelihood [32]. The EEG signal received at each electrode is correlated to the signals at other electrodes in space and time. In contrast to this traditional approach, the quantum-like structure

**Table 2. Mean of quantum potential for participants groups.**

| | Mean | SD | N | p-values | | | | |
| --- | --- | --- | --- | --- | --- | --- | --- | --- |
| | | | | **Control** | **Depression** | **Schizophrenia** | **AD** | **MCI** |
| Control | 4.149 | 0.02656 | 95 | 1 | $3.0088e^{-234}$ | $1.1298e^{-176}$ | $5.21198e^{-09}$ | $4.76607e^{-14}$ |
| Depression | 4.24 | 0.03509 | 28 | $3.0088e^{-234}$ | 1 | $1.91703e^{-10}$ | $3.1218e^{-251}$ | $1.1569e^{-173}$ |
| Schizophrenia | 4.256 | 0.02983 | 42 | $1.1298e^{-176}$ | $1.91703e^{-10}$ | 1 | $5.7517e^{-197}$ | $2.9676e^{-122}$ |
| AD | 4.166 | 0.03236 | 40 | $5.21198e^{-09}$ | $3.1218e^{-251}$ | $5.7517e^{-197}$ | 1 | $9.47087e^{-34}$ |
| MCI | 4.136 | 0.02891 | 25 | $4.76607e^{-14}$ | $1.1569e^{-173}$ | $2.9676e^{-122}$ | $9.47087e^{-34}$ | 1 |

**Table 3. Variability of quantum potential for participants groups.**

|  | Variability | SD | N | p-values | | | | |
|---|---|---|---|---|---|---|---|---|
|  |  |  |  | Control | Depression | Schizophrenia | AD | MCI |
| Control | 0.1635 | 0.01475 | 95 | 1 | 0 | 0 | $7.4345e^{-05}$ | $4.0350e^{-12}$ |
| Depression | 0.3854 | 0.04649 | 28 | 0 | 1 | $5.7772e^{-67}$ | 0 | 0 |
| Schizophrenia | 0.4608 | 0.05635 | 42 | 0 | $5.7772e^{-67}$ | 1 | 0 | 0 |
| AD | 0.157 | 0.02644 | 40 | 7.4345e-05 | 0 | 0 | 1 | $9.96e^{-02}$ |
| MCI | 0.1541 | 0.01998 | 25 | $4.0350e^{-12}$ | 0 | 0 | $9.9619e^{-02}$ | 1 |

AD–Alzheimer's Disease; MCI–mild cognitive impairment; SD–standard deviation

of the brain, according to de Broglie-Bohm, allows quantifying connectivity as a function of space and time as well as of instantaneous quantum effects in space (non-locality) [27]. We examined both instantaneous (non-local) and non-instantaneous (local) interactions between the QP ($QP_{elec,t,participant}$) of each participant's EEG electrodes. In order to identify the non-instantaneous (local) interactions between these QPs, we examined for each participant, the maximal absolute correlation coefficient between the participant's electrodes over the whole recording time. This represents the *local* non-instantaneous effect between pairs of electrodes. Results are shown as heat maps of the mean of all participants' cross correlation (Fig 2A and 2D; Table 4).

To characterize the temporal relationship between the QP ($QP_{elec,t,participant}$) of two electrodes, we identified the time lag ($t_{lag}$) between maximal correlation coefficients of each of the two EEG electrodes (Fig 2B, 2E; Table 5). The *non-local* (lag = 0) instantaneous effect of the absolute correlation coefficient between each pair of electrodes were significantly different between all participant groups except between control and AD group (Fig 2C, 2F; Table 6).

## QP power spectrum analysis of $Q_{elec}$ values

Power spectral density (PSD) has been used to determine levels of brain activity in EEG recordings in order to assess the power of each frequency observed in various states of consciousness [33, 34]. Next PSD analysis was used to evaluate in the various clinical groups, the QP power over a range of frequencies, as described in the methods. A normalized PSD of $QP_{elec,t,participant}$ was calculated using fast Fourier transformation (FFT), for each participant group (Fig 3). The QP power discriminated between the participant groups. Namely, each group showed a distinct pattern as for example, while in participants with depression no power differences were detected at various frequencies, the control group showed significant differences ($p<0.05$; ANOVA) in all except one ($2e^{-1}$ vs. $2e^{-2}$) frequency comparison. Most interestingly, when comparing participant groups to each other (multiple comparisons ANOVA), the power of the $QP_{elec,t,participant}$ of the depression and schizophrenia groups differed significantly when compared to all other participant groups at any frequency while no power differences were found at any frequency between control, AD, and MCI groups (Table 7). Comparing the PSD across all frequencies and participant groups by normalizing the maximum power of each frequency band resulted in significant differences between participants with depression ($0.879 \pm 0.08$) or schizophrenia ($0.688 \pm 0.08$) versus all other groups ($p<0.0001$ for both) while participants with AD ($0.456 \pm 0.05$), MCI ($0.478 \pm 0.05$), and control participants ($0.463 \pm 0.04$) did not differ from each other ($p>0.05$; Table 8; Fig 3).

## EEG QP accuracy in differentiating neuropsychiatric diseases from controls

In order to employ the method described above to classify individual participants by their clinical neuropsychiatric diagnosis we combined the mean values and the dynamic variability of the EEG QP. It was shown earlier that the value of the EEG QP represented by $Qme_{t,group}$ as

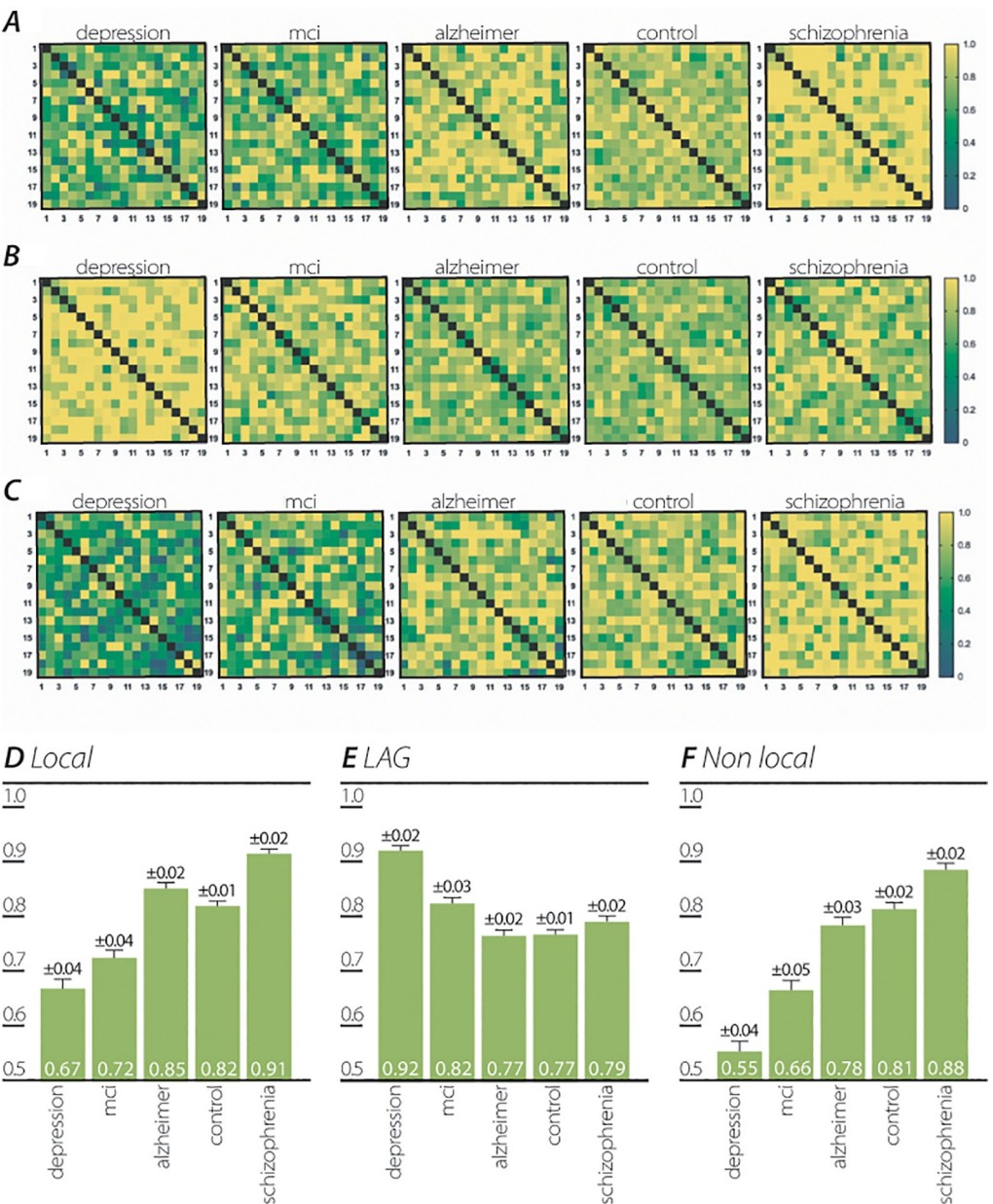

**Fig 2. Cross correlation of time series QP between EEG electrodes.** (A)Heatmaps of maximum correlation coefficients between each pair of electrodes QP ($QP_{elec,t, participant}$) time series. Each correlation coefficient was normalized to the maximum correlation coefficient of all patient groups. (B) Heatmaps of temporal difference (lag) of maximum correlation coefficient between each pair of electrodes QP ($QP_{elec,t,participant}$) time series. Each temporal difference was normalized to the maximum temporal difference of all patient groups. (C) Heatmaps of instantaneous correlation coefficients (lag = 0) between each pair of electrodes QP ($QP_{elec,t,participant}$) time series. Each instantaneous correlation coefficient was normalized to the maximum instantaneous correlation coefficient of all patient groups. (D) Mean maximum correlation coefficients. (E) Mean of temporal difference (lag) of maximum correlation coefficient. (F) Mean of instantaneous correlation coefficients (lag = 0). Error bars indicate standard error of mean (SEM).

well as the dynamics of EEG QP embodied by the results from the spectral analysis ($<SP_{elec}>_{participant}>_{window}$) can distinguish between the participant groups and healthy controls. Thus, a combined score of the mean and variability of the EEG based QP (qpmvs) was calculated for each participant (see methods above). The accuracy of the identification of

**Table 4. Cross correlation analysis between participant's electrodes–local.**

| | Variability | SD | N | p-values | | | | |
| --- | --- | --- | --- | --- | --- | --- | --- | --- |
| | | | | Control | Depression | Schizophrenia | AD | MCI |
| Depression | 0.6686 | 0.2091 | 28 | 1 | 0.0170 | 0.001<< | 0.001<< | 0.001<< |
| MCI | 0.7229 | 0.1916 | 25 | 0.0170 | 1 | 0.001<< | 0.001<< | 0.001<< |
| AD | 0.8492 | 0.1473 | 40 | 0.001<< | 0.001<< | 1 | 0.4266 | 0.0027 |
| Control | 0.8192 | 0.1192 | 95 | 0.001<< | 0.001<< | 0.4266 | 1 | 0.001<< |
| Schizophrenia | 0.9129 | 0.1213 | 42 | 0.001<< | 0.001<< | 0.0027 | 0.001<< | 1 |

ANOVA–multiple comparison; AD–Alzheimer's Disease; MCI–mild cognitive impairment; SD–standard deviation

participants' diagnoses (healthy, depression, schizophrenia, AD and MCI) that was based solely on the qpmvs derived from routine EEG recordings, was examined by ROC analysis and the resulting AUC values. ROC analysis showed high accuracy for the qpmvs when comparing control participants to participants with schizophrenia (AUC = 0.8981± 0.028, 95% CI: (0.8426–0.9535), p<0.0001), depression (AUC = 0.9033± 0.028, 95% CI: (0.8479–0.9586), p<0.0001), AD (AUC = 0.9143± 0.042, 95% CI: (0.8312–0.9974), p<0.0001), and MCI (AUC = 0.8309± 0.06228, 95% CI: (0.7088–0.9529), p<0.0001; Fig 4).

## EEG QP accuracy in differentiating between the neuropsychiatric and neurocognitive disease groups

The qpmvs accuracy was tested for its ability to discriminate between pairs of neuropsychiatric and neurocognitive groups. High accuracy was obtained in differentiating schizophrenia from depression (AUC = 0.8992± 0.055, 95% CI: (0.7910–1.000), p<0.0001), AD (AUC = 0.8762± 0.048, 95% CI: (0.7818–0.9706), p<0.0001) or MCI (AUC = 0.8914± 0.059, 95% CI: (0.7756–1.000), p<0.0001), as well as depression from AD (AUC = 0.8777± 0.048, 95% CI: (0.7828–0.9726), p<0.0001), or MCI (AUC = 0.8929± 0.058, 95% CI: (0.7781–1.000), p<0.0001; Fig 5). These results reveal that, participants with neuropsychiatric or neurocognitive disorders can be differentiated with a high level of accuracy not only from the healthy controls but also from each other, pointing to the qpmvs as a potentially useful diagnostic marker for differentiating between the hereby tested diagnoses.

## EEG QP accuracy in differentiating between the neurocognitive disease groups (AD and MCI)

A suboptimal accuracy was found in the differentiation between the two neurocognitive groups, AD vs. MCI (AUC = 0.7660± 0.06559, 95% CI: (0.6374–0.8946), p = 0.0003). In a

**Table 5. Cross correlation analysis between participant's electrodes–lag.**

| | Variability | SD | N | p-values | | | | |
| --- | --- | --- | --- | --- | --- | --- | --- | --- |
| | | | | Control | Depression | Schizophrenia | AD | MCI |
| Depression | 0.9206 | 0.1131 | 28 | 1 | 0.001<< | 0.001<< | 0.001<< | 0.001<< |
| MCI | 0.8234 | 0.1583 | 25 | 0.001<< | 1 | 0.0013 | 0.0017 | 0.1957 |
| AD | 0.7654 | 0.1419 | 40 | 0.001<< | 0.0013 | 1 | 0.9999> | 0.4510 |
| Control | 0.7665 | 0.127 | 95 | 0.001<< | 0.0017 | 0.9999> | 1 | 0.4980 |
| Schizophrenia | 0.7907 | 0.154 | 42 | 0.001<< | 0.1957 | 0.4510 | 0.4980 | 1 |

ANOVA–multiple comparison; AD–Alzheimer's Disease; MCI–mild cognitive impairment; SD–standard deviation

**Table 6. Cross correlation analysis between participant's electrodes–non-local.**

| | Variability | SD | N | | | p-values | | |
|---|---|---|---|---|---|---|---|---|
| | | | | Control | Depression | Schizophrenia | AD | MCI |
| Depression | 0.5535 | 0.2356 | 28 | 1 | 0.001<< | 0.001<< | 0.001<< | 0.001<< |
| MCI | 0.6639 | 0.2357 | 25 | 0.001<< | 1 | 0.001<< | 0.001<< | 0.001<< |
| AD | 0.7822 | 0.1938 | 40 | 0.001<< | 0.001<< | 1 | 0.6427 | 0.001<< |
| Control | 0.8113 | 0.1514 | 95 | 0.001<< | 0.001<< | 0.6427 | 1 | 0.0071 |
| Schizophrenia | 0.8826 | 0.141 | 42 | 0.001<< | 0.001<< | 0.001<< | 0.0071 | 1 |

ANOVA–multiple comparison; AD–Alzheimer's Disease; MCI–mild cognitive impairment; SD–standard deviation

subsequent analysis, the MCI group was divided into two subgroups, one that showed a stable disease course (stbMCI) and another with a deteriorating disease course (detMCI). Participants in the two groups did not differ in age or baseline cognitive testing scores and all had been clinically classified at baseline, as MCI with no indication to predicting their clinical course (stbMCI versus detMCI: age, 74.3± 4.5 years versus 73.3± 5.6 years, p = 0.72; MMSE scores, 28.6±1.2 versus 27.2±1.6, p = 0.09). Using the qpmvs, the stbMCI and detMCI groups were differentiated with a fair level of accuracy (AUC = 0.9815± 0.029, 95% CI: (0.9245–1.000), p = 0.0022). In addition, the stbMCI subgroup could be clearly distinguished from the AD group (AUC = 0.950± 0.044, 95% CI: (0.8622–1.000), p<0.0004), while the detMCI group was indistinguishable from the AD group (AUC = 0.533± 0.112, 95% CI: (0.3142–0.7524), p = 0.756; Fig 6).

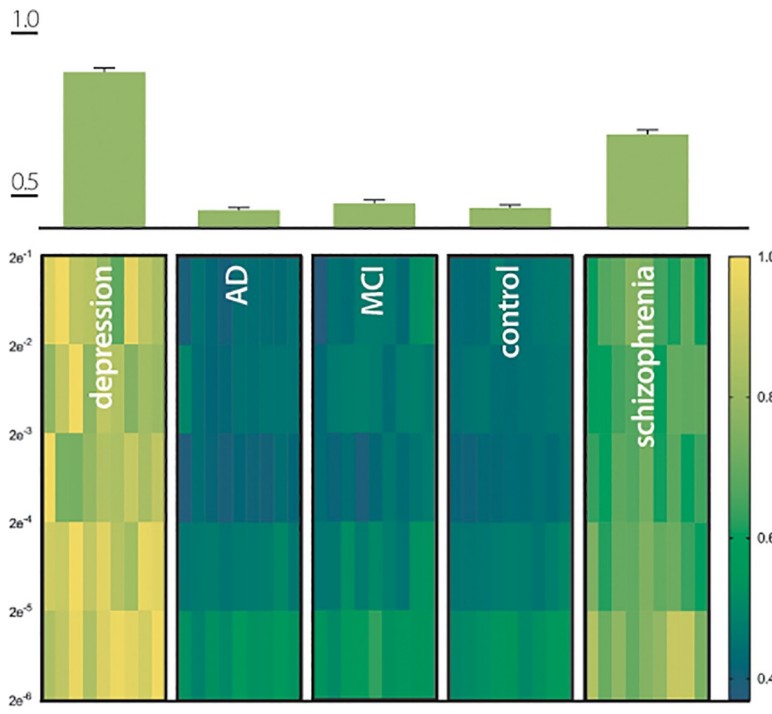

**Fig 3. Power spectral density (PSD) of QP for different patient groups.** (A) PSD for each $QP_{electrode}$ with frequency band of $2^{-n}$ (n = 1. . .5) and a window of 64s with 32s overlap, normalized values according to methods. (B) Graphic depiction of the mean and SEM of the normalized PSD of each patient group.

**Table 7. Intragroup comparison of QP power spectrum frequencies of QP$_{electrode}$ values.**

| frequency bands | participants group | | | | |
|---|---|---|---|---|---|
| | **Depression** | **AD** | **MCI** | **Control** | **Schizophrenia** |
| 2e- 1 vs. 2e$^{-2}$ | 0.9068 | 0.2380 | 0.9971 | 0.8651 | 0.9944 |
| 2e- 1 vs. 2e$^{-3}$ | 0.9078 | 0.4516 | 0.3570 | 0.0004 | 0.9691 |
| 2e- 1 vs. 2e$^{-4}$ | 0.7275 | 0.0003 | 0.1932 | 0.0128 | 0.9468 |
| 2e- 1 vs. 2e$^{-5}$ | 0.5859 | <0.0001 | <0.0001 | <0.0001 | 0.0003 |
| 2e- 2 vs. 2e$^{-3}$ | >0.9999 | 0.0050 | 0.2056 | 0.0053 | 0.9993 |
| 2e- 2 vs. 2e$^{-4}$ | 0.2413 | 0.0711 | 0.3390 | 0.0009 | 0.7875 |
| 2e- 2 vs. 2e$^{-5}$ | 0.1584 | <0.0001 | <0.0001 | <0.0001 | <0.0001 |
| 2e- 3 vs. 2e$^{-4}$ | 0.2424 | <0.0001 | 0.0023 | <0.0001 | 0.6543 |
| 2e- 3 vs. 2e$^{-5}$ | 0.1592 | <0.0001 | <0.0001 | <0.0001 | <0.0001 |
| 2e- 4 vs. 2e$^{-5}$ | 0.9993 | <0.0001 | 0.0005 | <0.0001 | 0.0020 |

ANOVA—multiple comparison; AD—Alzheimer's Disease; MCI—mild cognitive impairment

## Discussion

Multiple neuropsychiatric diseases including depression, schizophrenia, and neurocognitive disorders (AD and MCI) can be differentiated by extracting characteristic information patterns from dendrograms that present the hierarchical, treelike structure of information processing in the brain, encoded as $p$-adic numbers. This study attempted to use an EEG pattern as a marker of an individual's brain state [27].

In the model used in the current study, QP was defined as information extracted by means of $p$-adic encoding of the dendrogram representing the hierarchic, integrated and non-local structure of each participant's EEG. As described above the value of each participant group's QP was then quantified, exposing a distinct and significant differentiation among the groups (mean of the $Qme_{t,group}$). Furthermore, the variability of QP among participants in each group, indicated by $Qstd_{t,group}$, differentiated between the groups and represented a variability factor for each of them.

As information processing in the brain is assumed to be non-local and resulting from the integration of neuronal activity in various areas of the brain [35, 36], the quantum-like

**Table 8. Between group comparison of the QP power spectrum of QP$_{electrode}$ values.**

| | frequency bands | | | | | all frequencies |
|---|---|---|---|---|---|---|
| | **2e$^{-1}$** | **2e$^{-2}$** | **2e$^{-3}$** | **2e$^{-4}$** | **2e$^{-5}$** | **2e$^{-1}$ to 2e$^{-5}$** |
| depression vs. AD | <0.0001 | <0.0001 | <0.0001 | <0.0001 | <0.0001 | <0.0001 |
| depression vs. MCI | <0.0001 | <0.0001 | <0.0001 | <0.0001 | <0.0001 | <0.0001 |
| depression vs. control | <0.0001 | <0.0001 | <0.0001 | <0.0001 | <0.0001 | <0.0001 |
| depression vs. schizophrenia | <0.0001 | <0.0001 | <0.0001 | <0.0001 | <0.0001 | <0.0001 |
| AD vs. MCI | 0.7773 | 0.9072 | 0.8663 | 0.8274 | 0.8680 | 0.1633 |
| AD vs. control | 0.9094 | >0.9999 | 0.9737 | >0.9999 | >0.9999 | 0.9429 |
| AD vs. schizophrenia | <0.0001 | <0.0001 | <0.0001 | <0.0001 | <0.0001 | <0.0001 |
| MCI vs. control | 0.9983 | 0.9075 | 0.9959 | 0.8219 | 0.8647 | 0.5232 |
| MCI vs. schizophrenia | <0.0001 | <0.0001 | <0.0001 | <0.0001 | <0.0001 | <0.0001 |
| control vs. schizophrenia | <0.0001 | <0.0001 | <0.0001 | <0.0001 | <0.0001 | <0.0001 |

ANOVA—multiple comparison; AD—Alzheimer's Disease; MCI—mild cognitive impairment

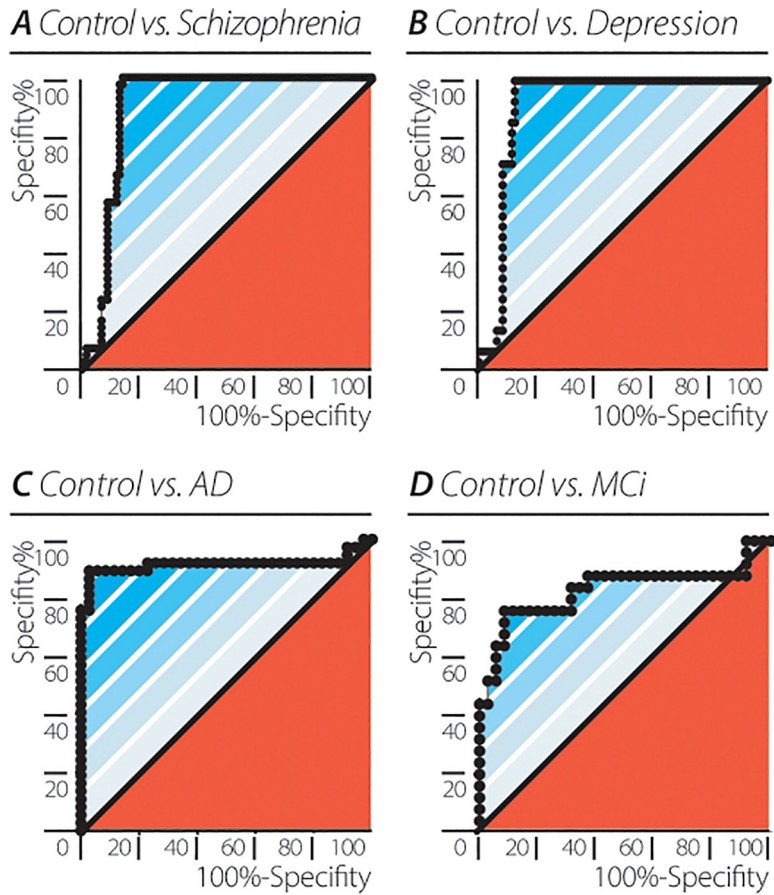

**Fig 4. Accuracy of the EEG based quantum potential mean and variability score (qpmvs) in differentiating neuro-psychiatric patient groups from healthy controls.** Receiver operating characteristic (ROC) curves for (A) control vs. schizophrenia (AUC = 0.8981± 0.028, 95% CI: (0.8426–0.9535), p = <0.0001). (B) control vs. depression (AUC = 0.9033± 0.028, 95% CI: (0.8479–0.9586), p = <0.0001). (C) control vs. AD (AUC = 0.9143± 0.042, 95% CI: (0.8312–0.9974), p = <0.0001). (D) control vs. MCI (AUC = 0.8309± 0.06228, 95% CI: (0.7088–0.9529), p = <0.0001).

structure allows not only quantification of connectivity as a function of space and time (locality) but also of instantaneous quantum effects in space (non-locality) [27]. By quantifying the relationships between elements of the hierarchic structure derived from the EEG electrodes, it was possible to examine both instantaneous (non-local) and non-instantaneous (local) interactions between each participant's EEG electrode QP ($QP_{elec,t,participant}$), showing highly significant differences among participants with various neuropsychiatric and neurocognitive diagnoses. Thus, both instantaneous (non-local) and non-instantaneous (local) interactions between different parts of the brain are modulated to different degrees, by neuropsychiatric and neurocognitive disorders. Furthermore, the dynamic (temporal) transition from one EEG electrode-derived hierarchical structure to another is shown above. The power spectral analysis of the QP suggests that the transition pattern from one hierarchical structure to the other, is highly dependent on the clinical state of the participant, very stable across frequency bands and highly segregated regarding clinical diagnoses.

The accuracy of the combined, hierarchic, whole emergent brain function quantification represented by the value of the QP, in diagnosing neuropsychiatric and neurocognitive diseases, was evaluated using ROC analysis. Paired ROC analysis of healthy controls vs. participants with depression, schizophrenia, AD and MCI showed AUCs with extremely high values,

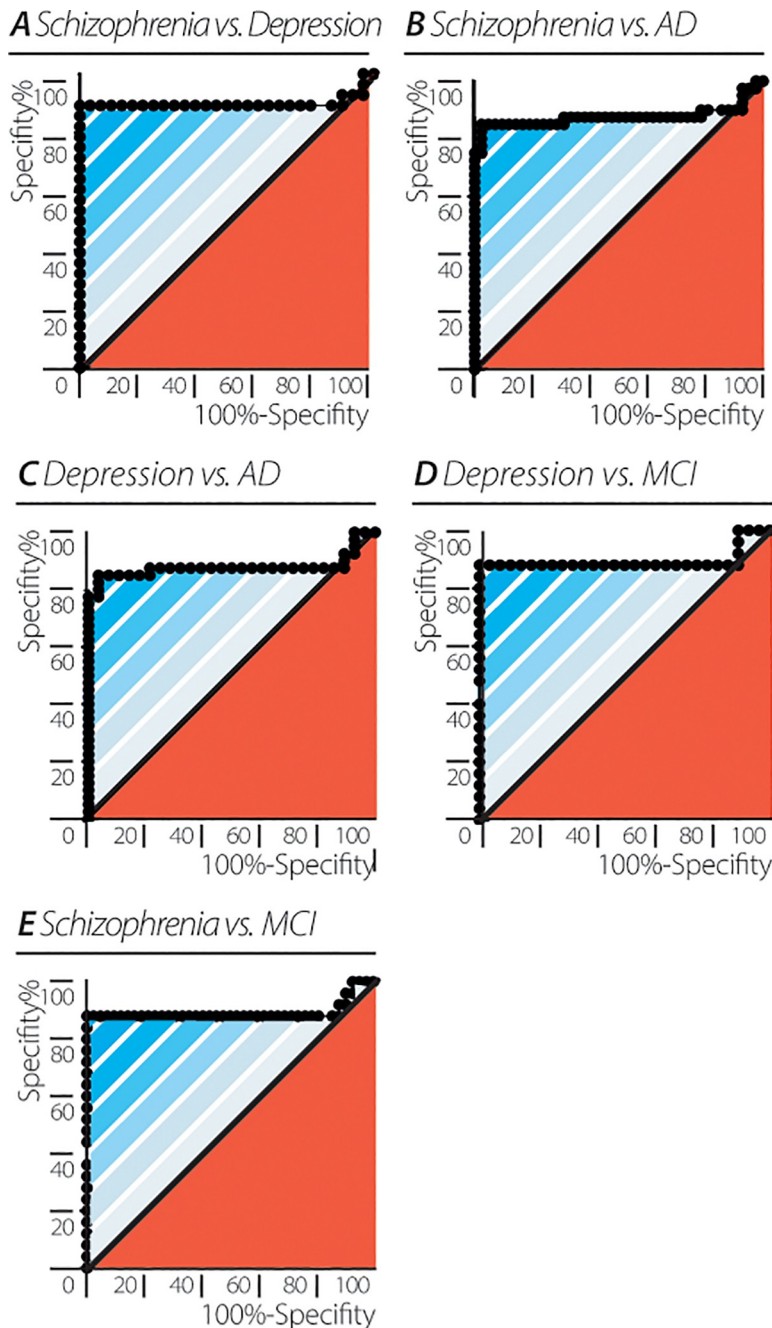

**Fig 5. Differentiating between neuro-psychiatric patient groups by EEG quantum potential mean and variability score (qpmvs).** Accuracy depicted as receiver operating characteristic (ROC) curves for (A) schizophrenia vs. depression (AUC = 0.8992± 0.055, 95% CI: (0.7910–1.000), p = <0.0001). (B) schizophrenia vs. AD (AUC = 0.8762± 0.048, 95% CI: (0.7818–0.9706), p = <0.0001). (C) schizophrenia vs. MCI (AUC = 0.8914± 0.059, 95% CI: (0.7756–1.000), p = <0.0001). (D) depression vs. AD (AUC = 0.8777± 0.048, 95% CI: (0.7828–0.9726), p = <0.0001). (E) depression vs. MCI (AUC = 0.8929± 0.058, 95% CI: (0.7781–1.000), p = <0.0001).

thus indicating that the qpmvs may be a useful tool for diagnosing the presence of neuropsychiatric or neurocognitive diseases. Furthermore, AUCs also showed high values, namely, high accuracy, in differentiating between schizophrenia or depression and cognitive decline (AD

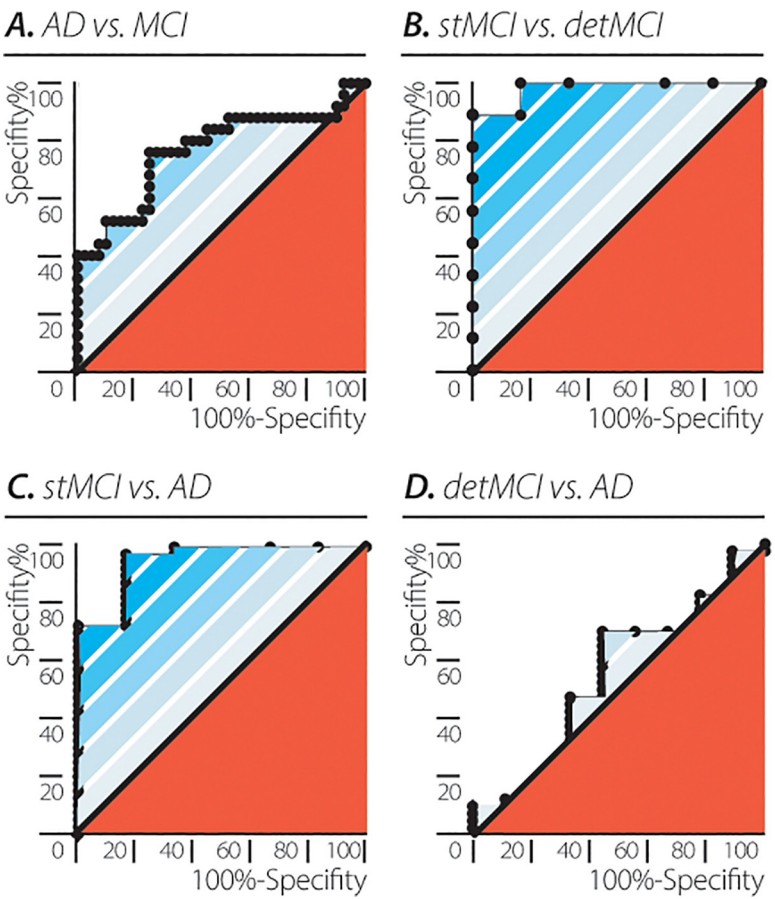

**Fig 6. Accuracy of differentiating between neurocognitive diseases using the EEG quantum potential mean and variability score (qpmvs).** Accuracy depicted as receiver operating characteristic (ROC) curves for (A) AD vs. MCI (AUC = 0.7660± 0.06559, 95% CI: (0.6374–0.8946), p = 0.0003). (B) stbMCI vs. detMCI (AUC = 0.9815± 0.029, 95% CI: (0.9245–1.000), p = 0.0022). (C) stbMCI vs. AD (AUC = 0.950± 0.044, 95% CI: (0.8622–1.000), p = <0.0004). (D) detMCI vs. AD (AUC = 0.533± 0.112, 95% CI: (0.3142–0.7524), p = 0.756).

and MCI). In addition, the algorithm used predicted with extremely high accuracy a stable disease course (stbMCI) or a progressively deteriorating course (detMCI) that would eventually lead to dementia (Fig 6). EEG analysis did not distinguish between detMCI and AD, despite participants with detMCI being diagnosed with MCI. These findings show the qpmvs to be a highly sensitive tool for predicting disease course and to be used as a biomarker for diagnosis in early disease stages of cognitive decline as well as possibly assisting in identifying the people at high risk of future cognitive deterioration. One limitation of our study is related to the heterogenic group of healthy controls. This group is comprised of patients which underwent routine EEG without a clear indication and no neuro-psychiatric disorders but might include those with unspecific headaches including migraine and tension-type headache or dizziness. Currently we cannot exclude an influence of those complaints on the qpmvs.

To the best of the authors' knowledge, this is the first medical diagnostic study suggesting the use of ultrametric analyses, closely coupled with the theory of *p*-adic numbers, and quantum theory. We use the formalism of quantum mechanics for modelling information processing in the brain, without consideration of quantum physical processes in it: our model is quantum-like, not genuine quantum [37–43]. Such models have already found numerous applications in psychology and decision making (see [19–21, 44, 45] and references herein).

But this is the first work on real medical diagnostic based on the quantum-like model. As such a new paradigm that does not involve frequency bands, regular spectral analysis, or feature extraction, solely based on routine EEG recordings without a specific research setting, is suggested. It is also expected for this combination of quantum theory with a hierarchical (non-local) treelike representation of information processing in the brain, to find novel applications in medical and cognitive sciences.

## Acknowledgments

Varda and Boaz Dotan for discussion; Michal Shor and Esia Tzaphnat for graphic design.

## Author Contributions

**Conceptualization:** Oded Shor, Andrei Khrennikov, Felix Benninger.

**Data curation:** Oded Shor, Amit Yaniv-Rosenfeld, Avi Valevski, Andrei Khrennikov, Felix Benninger.

**Formal analysis:** Oded Shor, Amit Yaniv-Rosenfeld, Andrei Khrennikov, Felix Benninger.

**Funding acquisition:** Andrei Khrennikov, Felix Benninger.

**Investigation:** Oded Shor, Andrei Khrennikov, Felix Benninger.

**Methodology:** Oded Shor, Avi Valevski, Andrei Khrennikov, Felix Benninger.

**Project administration:** Oded Shor, Andrei Khrennikov, Felix Benninger.

**Resources:** Oded Shor, Amir Glik, Avi Valevski, Andrei Khrennikov, Felix Benninger.

**Software:** Oded Shor, Andrei Khrennikov, Felix Benninger.

**Supervision:** Oded Shor, Abraham Weizman, Andrei Khrennikov, Felix Benninger.

**Validation:** Oded Shor, Andrei Khrennikov, Felix Benninger.

**Visualization:** Oded Shor, Andrei Khrennikov, Felix Benninger.

**Writing – original draft:** Oded Shor, Avi Valevski, Abraham Weizman, Andrei Khrennikov, Felix Benninger.

**Writing – review & editing:** Oded Shor, Avi Valevski, Abraham Weizman, Andrei Khrennikov, Felix Benninger.

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
