## [Decision Letter · Decision Letter 0]

15 Jun 2021

PONE-D-21-10465

EEG p-adic quantum potential accurately identifies depression, schizophrenia and cognitive decline

PLOS ONE

Dear Dr. Benninger,

Thank you for submitting your manuscript to PLOS ONE. After careful consideration, we feel that it has merit but does not fully meet PLOS ONE’s publication criteria as it currently stands. Therefore, we invite you to submit a revised version of the manuscript that addresses the points raised during the review process.

We look forward to receiving your revised manuscript.

Kind regards,

Giuseppe Vitiello

Academic Editor

PLOS ONE

2.Thank you for including your ethics statement: "The study adhered to rules and regulations of the Helsinki Declaration and was approved by the relevant IRBs (0275-20-RMC)."

b. Please provide additional details regarding participant consent. In the ethics statement in the Methods and online submission information, please ensure that you have specified (1) whether consent was informed and (2) what type you obtained (for instance, written or verbal, and if verbal, how it was documented and witnessed). If your study included minors, state whether you obtained consent from parents or guardians. If the need for consent was waived by the ethics committee, please include this information.

Additional Editor Comments (if provided):

Reviewers' comments:

Reviewer's Responses to Questions

**Comments to the Author**

1. Is the manuscript technically sound, and do the data support the conclusions?

Reviewer #1: Yes

Reviewer #2: Yes

2. Has the statistical analysis been performed appropriately and rigorously? 

Reviewer #1: Yes

Reviewer #2: I Don't Know

3. Have the authors made all data underlying the findings in their manuscript fully available?

Reviewer #1: Yes

Reviewer #2: No

4. Is the manuscript presented in an intelligible fashion and written in standard English?

Reviewer #1: No

Reviewer #2: Yes

5. Review Comments to the Author

Reviewer #1: The biological information transduction behaves quantum-like, therefore, it is well described and understood according to the principle and operations of quantum mechanism. The manuscript is a marvelous application of the research achievement of trying to understand the biological information processing in terms of quantum mechanism. I believe the success of this application in diagnosis of psychiatric patients should be widely open. Further, the success of discrimination between various types of such diseases is especially surprising, which can be used in research development to understand the detailed mechanism of the origin and development of such various diseases.

Minor concern is: “Does the control contain not only healthy people but also other brain patients than the psychiatric patients reported in this manuscript, such as brain cancer, migraine, and so on?” This is important at the practical application for the medical diagnosis of each individual patient. At least the authors should mention about this at the discussion section.

English is awkward at several places. Corrections by a native speaker should help the improvement.

Reviewer #2: The paper introduces new a quantum potential mean and variability score (qpmvs), to identify neuropsychiatric and neurocognitive disorders with high accuracy, based on EEG recordings. It is fascinating, highly interdisciplinary, paper involving quantum physics, neurophysiology, among other areas. My report is dedicated only to the physics/mahematics of the qpmvs. In my view, the description of the underlying physical model requires serious improvements. I suggest including an appendix explaining the model and its numerical implementation. Please see the attached file.

6. PLOS authors have the option to publish the peer review history of their article (what does this mean?). If published, this will include your full peer review and any attached files.

Reviewer #1: **Yes: **Ichiro Yamato

Reviewer #2: No

---

## [Author Response · Author response to Decision Letter 0]

10 Jul 2021

Response to Reviewers - EEG p-adic quantum potential accurately identifies depression, schizophrenia and cognitive decline

Dear Reviewers,

Thank you very much for your encouraging comments and your great help to improve the manuscript. According to your suggestions, we made appropriate changes highlighted in the manuscript with markups. 

We added a section in the discussion to comply with the constructive comments from reviewer number one. The inclusion of healthy control patients was discussed as limitation complying with the very well-placed comment (“One limitation of our study is related to the heterogenic group of healthy controls. This group is comprised of patients which underwent routine EEG without a clear indication and no neuro-psychiatric disorders but might include those with unspecific headaches including migraine and tension-type headache or dizziness. Currently we cannot exclude an influence of those complaints on the qpmvs.”

The manuscript was also edited to improve language. 

We would like to thank the second reviewer for his constructive remark regarding the use of the same symbol (p) for two different mathematical entities. This might have caused a disturbance by the reader. As the letter p is standard for both areas, statistics and number theory we decided to use a bold and italic fond when referring to p-adic and now expect a clearer distinguishability between these two notations. As the comment on the use of 2-adic numbers throughout the paper, we added a remark (parentheses 2 last lines of page 4). We would like to preserve the presentation for an arbitrary p, since it can be useful in some applications. In this context, “Q_p” for the field of p-adic numbers, was exchanged with both letters as capital “QP”. 

We further re-wrote the methods part regarding the 2-adic quantum potential calculation sub-section and the section describing the quantum potential mean and variability score (qpmvs). To improve the understanding regarding the 2-adic quantum potential calculation, we added as suggested a series of algorithm steps with detailed explanations regarding each step and improved the mathematical terms in each equation. Moreover, we defined rigidly the averages and standard deviation terms. These changes have been made to meet with the remarks #2 and #4 of reviewer number 2. 

We also like to comment on the seconds reviewer remark #3 (“The computation of the quantum potential requires the numerical computation of derivatives of functions having jumping-type singularities, these derivatives naturally contain Dirac delta functions”). As can be seen from the changes in the 2-adic quantum potential calculation sub-section, step 9, equations 5-7, the numerical calculation do not involve such Dirac delta functions. Moreover, the meaning of the QP function is explained in step 9 as “Thus, the ambiguous quantum potential notion becomes in our framework quite trivially a score or measure of hierarchical topology”. This meaning of the QP is a direct implication of our numerical 2-adic quantum potential computation but can be extended to the non-numerical cases. 

The data availability was updated and improved and the original data files are anonymized and available as a three-dimensional matrix as a matlab file at https://datadryad.org/stash/share/AWmC0-Afzx29cOkYDXQ6y2-7HF4GBvG-J-9i8hDQZsw which has been added to the manuscript.

---

## [Decision Letter · Decision Letter 1]

19 Jul 2021

EEG p-adic quantum potential accurately identifies depression, schizophrenia and cognitive decline

PONE-D-21-10465R1

Dear Dr. Benninger,

We’re pleased to inform you that your manuscript has been judged scientifically suitable for publication and will be formally accepted for publication once it meets all outstanding technical requirements.

Kind regards,

Giuseppe Vitiello

Academic Editor

PLOS ONE

Additional Editor Comments (optional):

Dear Dr. Felix Benninger,

it is a pleasure to communicate you that your paper has been accepted to be published on PLOS ONE.

Best regards

Giuseppe Vitiello

Reviewers' comments:

Reviewer's Responses to Questions

**Comments to the Author**

1. If the authors have adequately addressed your comments raised in a previous round of review and you feel that this manuscript is now acceptable for publication, you may indicate that here to bypass the “Comments to the Author” section, enter your conflict of interest statement in the “Confidential to Editor” section, and submit your "Accept" recommendation.

Reviewer #1: All comments have been addressed

2. Is the manuscript technically sound, and do the data support the conclusions?

Reviewer #1: (No Response)

3. Has the statistical analysis been performed appropriately and rigorously? 

Reviewer #1: (No Response)

4. Have the authors made all data underlying the findings in their manuscript fully available?

Reviewer #1: (No Response)

5. Is the manuscript presented in an intelligible fashion and written in standard English?

Reviewer #1: (No Response)

6. Review Comments to the Author

Reviewer #1: You addressed adequately to my concern. Thank you for your providing a nice diagnostic analysis method for psychological diseases. I hope you continue investigation further on the control patients to discriminate clearly the psychological diseases from other kinds of brain diseases. Looking forward to reading your further reports.

I suggest you to be more careful about simple English errors such as: Introduction section, 2nd paragraph, Baring -> Bearing; Method section, 2-adic part, 4. normalized -> Normalized; time step, t, normalized -> time step, t, were normalized; and several others; Fig.1 legend, 3rd line, Qmet,group the for comparison -> Qmet,group for comparison.

7. PLOS authors have the option to publish the peer review history of their article (what does this mean?). If published, this will include your full peer review and any attached files.

Reviewer #1: **Yes: **Ichiro Yamato

---

## [Editor Report · Acceptance letter]

27 Jul 2021

PONE-D-21-10465R1 

EEG *p*-adic quantum potential accurately identifies depression, schizophrenia and cognitive decline 

Dear Dr. Benninger:

I'm pleased to inform you that your manuscript has been deemed suitable for publication in PLOS ONE. Congratulations! Your manuscript is now with our production department. 

Kind regards, 

on behalf of

Dr. Giuseppe Vitiello 

Academic Editor

PLOS ONE